# Predictive Value of MR-proADM in the Risk Stratification and in the Adequate Care Setting of COVID-19 Patients Assessed at the Triage of the Emergency Department

**DOI:** 10.3390/diagnostics12081971

**Published:** 2022-08-15

**Authors:** Marilena Minieri, Vito N. Di Lecce, Maria Stella Lia, Massimo Maurici, Francesca Leonardis, Susanna Longo, Luca Colangeli, Carla Paganelli, Stefania Levantesi, Alessandro Terrinoni, Vincenzo Malagnino, Domenico J. Brunetti, Alfredo Giovannelli, Massimo Pieri, Marco Ciotti, Cartesio D’Agostini, Mariachiara Gabriele, Sergio Bernardini, Jacopo M. Legramante

**Affiliations:** 1Department of Experimental Medicine, University of Rome Tor Vergata, 00133 Rome, Italy; 2Laboratory Medicine Unit, Tor Vergata University Hospital, 00133 Rome, Italy; 3Emergency Department, Tor Vergata University Hospital, 00133 Rome, Italy; 4Department of Biomedicine and Prevention, University of Rome Tor Vergata, 00133 Rome, Italy; 5Department of Surgical Sciences, University of Rome Tor Vergata, 00133 Rome, Italy; 6Intensive Care Unit, Emergency Department, Tor Vergata University Hospital, 00133 Rome, Italy; 7Department of Systems Medicine, University of Rome Tor Vergata, 00133 Rome, Italy; 8Infectious Disease Unit, Tor Vergata University Hospital, 00133 Rome, Italy; 9Anaesthesia and Intensive Care Unit, Tor Vergata University Hospital, 00133 Rome, Italy; 10Virology Unit, Tor Vergata University Hospital, 00133 Rome, Italy; 11Clinical Microbiology Unit, Tor Vergata University Hospital, 00133 Rome, Italy; 12Respiratory Medicine Unit, Tor Vergata University Hospital, 00133 Rome, Italy

**Keywords:** emergency department, triage, COVID-19 biomarkers, mid-regional proadrenomedullin

## Abstract

In the past two pandemic years, Emergency Departments (ED) have been overrun with COVID-19-suspicious patients. Some data on the role played by laboratory biomarkers in the early risk stratification of COVID-19 patients have been recently published. The aim of this study is to assess the potential role of the new biomarker mid-regional proadrenomedullin (MR-proADM) in stratifying the in-hospital mortality risk of COVID-19 patients at the triage. A further goal of the present study is to evaluate whether MR-proADM together with other biochemical markers could play a key role in assessing the correct care level of these patients. Data from 321 consecutive patients admitted to the triage of the ED with a COVID-19 infection were analyzed. Epidemiological; demographic; clinical; laboratory; and outcome data were assessed. All the biomarkers analyzed showed an important role in predicting mortality. In particular, an increase of MR-proADM level at ED admission was independently associated with a threefold higher risk of IMV. MR-proADM showed greater ROC curves and AUC when compared to other laboratory biomarkers for the primary endpoint such as in-hospital mortality, except for CRP. This study shows that MR-proADM seems to be particularly effective for early predicting mortality and the need of ventilation in COVID-19 patients admitted to the ED.

## 1. Introduction

The severe acute respiratory syndrome coronavirus 2 (SARS-CoV-2) pandemic has spread worldwide and reached catastrophic proportions in the past two years. The SARS-CoV-2 infection has been named coronavirus disease 2019 (COVID-19). The World Health Organization (WHO) declared the SARS-CoV-2 infection a ‘Public Health Emergency of International Concern’ due to its rapid transmission among humans. To date, there have been more than 539 million cases worldwide and more than 6 million deaths. SARS-CoV-2 infection presents a wide clinical spectrum ranging from asymptomatic infection to mild upper respiratory tract illness or severe interstitial pneumonia with respiratory failure [1].

During the COVID-19 pandemic, the high number of hospital admissions complicated patient management, highlighting the weakness of our health system, resulting in an increase of the mortality rate. Furthermore, the lack of specific clinical features of COVID-19 pneumonia complicated the differential diagnosis from other forms of severe pneumonia [1].

The lack of immediate results from the current microbiological tests to confirm COVID-19, coupled with the reported suboptimal sensitivity of swab tests by real-time reverse transcription-polymerase chain reaction (RT-PCR) assays, made the situation worse [2]. Consequently, Emergency Departments have been overwhelmed by COVID-19-suspicious patients, inducing the need to stratify the patient risk immediately upon entering the triage. Since predicting the course of this disease at symptom onset is difficult and very often clinical conditions tends to worsen abruptly, prognostic tools and/or biochemical markers have been fundamental to address patients through the right clinical pathway in the ED. However, although several laboratory biomarkers have been so far identified to diagnose COVID-19 pneumonia more rapidly, to date, there are no data on biomarkers with high specificity and sensibility able to early stratify the mortality risk of patients affected by viral pneumonia [3].

C-reactive protein (CRP) has been one of the most used biomarkers to assess the evolution of COVID-19 inflammatory processes, even though its use is limited by a low sensitivity for community-acquired pneumonia (CAP). While a high CRP value (>100 mg/L) can indicate a severe bacterial infection, lower values are common in both viral infections and noninfectious diseases [4].

Another biomarker evaluated in COVID-19 patients has been procalcitonin (PCT), since it has an important role in detecting a bacterial superinfection to manage antibiotic therapy [5]. As is known, PCT can predict microbial etiology in pneumonia [6]. On the other hand, in patients with a high pneumonia severity index (PSI classes III-V), PCT has proven to be a good prognostic marker rather than a diagnostic marker [7].

Procalcitonin, CRP and white blood cell count have been shown to be significantly higher in CAP patients with a typical bacterial etiology as compared to cases in which the pathogen was represented by an atypical bacterium or by a virus [8].

The mid-regional proadrenomedullin (MR-proADM) is the precursor molecule of adrenomedullin (ADM). Unlike ADM, which is unstable and characterized by a short half-life, MR-proADM is more stable and appears promising as biomarker for detecting endothelial dysfunction, therefore, predicting severity and long-term adverse outcomes in CAP. Christ-Crain et al. [9] showed that the level of MR-proADM increased with the severity of CAP, in contrast to CRP levels and leukocytes. In the study by Valenzuela Sanchez et al. [10], MR-proADM predicted unfavorable outcomes in patients with pneumonia caused by influenza virus. In addition, MR-proADM can be used to stratify the clinical risk in patients affected by CAP [11]. Determination of MR-proADM level within 6 h of arrival to the hospital has prognostic value. It has also been reported that MR-proADM obtained within 6 h from arrival at the hospital has considerable prognostic value, irrespective of the causal agent of CAP, and in association with PSI and CURB-65 scores, improves prognostic accuracy [11,12].

In the presence of an alteration of the microcirculatory integrity due to an endothelium damage with consequent capillary leak, as observed in sepsis, MR-proADM plasma concentrations tend to increase [13].

Accordingly, Hupf et al. [14] have recently reported significantly higher adrenomedullin RNA blood expression in patients with severe COVID-19 vs. patients with a mild disease.

It has been hypothesized by Li et al. [15] that the integrity of the epithelial-endothelial barrier is severely damaged in critical ill patients with COVID-19 pneumonia, introducing the concept of “viral sepsis”. Considering this pathogenetic mechanism, several recent studies reported an increased level of proinflammatory cytokines and chemokines such as tumor necrosis factor-α (TNF-α) and interleukin-6 (IL-6) in COVID-19 patients [16,17] suggesting that this cytokine storm might have a critical role in the evolution of SARS-CoV-2 infection [18].

The predictive value of MR-proADM in patients with COVID-19 pneumonia has been recently reported [19,20]. Our group showed that MR-proADM might have a predictive value in the early risk stratification of patients with COVID-19 infection at the triage in the Emergency Department [21].

Therefore, the aim of this study is to evaluate more in detail if the laboratory biomarkers can play a key role in predicting the correct care level of these patients, contributing to optimizing the hospital resources and helping the emergency physician in the decision about the adequate setting of care of patients at the entry to the Emergency Department.

## 2. Materials and Methods

### 2.1. Study Design

The present study has an observational, retrospective single-center design. Data from 321 consecutive patients admitted to the Emergency Department of the University Hospital Tor Vergata (Rome, Italy) from April to December 2020 with a confirmed COVID-19 infection were analyzed. A diagnosis of COVID-19 was made by a positive real-time reverse transcription polymerase chain reaction (RT-PCR) taken from nasopharyngeal swabs and through radiological imaging, where indicated, in accordance with WHO interim guidelines. Adult patients aged >18 years with a positive swab test were enrolled.

The demographic, epidemiological and clinical data were obtained from the electronic clinical records.

Patients underwent chest X-rays or computed tomography (CT) scans based on the physician’s clinical assessment, and these data were further revised by the Emergency Department’s radiologist. Blood culture, sputum, urine, bronchial aspirate and/or bronchoalveolar samples were analyzed when deemed necessary.

The final diagnosis was considered as that provided by the ED physician. A patient follow-up was performed up to 45 days.

The design of the study was evaluated and approved by the local Ethics Committee at Tor Vergata University Hospital (approval number 87/20) and was carried out according to the Declaration of Helsinki. Written informed consent was waived because of the rapid spread of this infectious disease.

### 2.2. Blood Sample Collection

Blood samples were collected at the triage admission. Upon arrival at the laboratory, blood samples were centrifuged at 4500× *g* for 5 min to obtain serum or plasma samples.

Blood examinations were for mid-regional proadrenomedullin (MR-proADM), C-reactive protein (CRP), procalcitonin (PCT), D-dimer, lactate dehydrogenase (LDH).

CRP (normality range 0.01–5.0 mg/L) and LDH (normality range 125–220 IU/L) levels were measured in serum using an Abbott ARCHITECT c16000 (Abbott, Chicago, IL, USA) clinical chemistry analyzer. PCT (normality range 0.01–0.50 ng/mL; cut-off for suspected infection 0.50 ng/mL) was detected in serum with a BRAHMS PCT chemiluminescent microparticle immunoassay (CMIA) by an Abbott ARCHITECT i2000SR instrument. MR-proADM (normality range 0.05–0.55 nmol/L) was measured using a time-resolved amplified cryptate emission assay on EDTA plasma samples (TRACE BRAHMS MR-proADM Kryptor, BRAHMS AG, Hennigsdorf, Germany). D-dimer values were obtained by an ACL TOP 700 instrument (Instrumentation Laboratory Company, Werfen, Bedford, MA, USA).

### 2.3. Statistical Analysis

The primary endpoint was the overall in-hospital mortality; the secondary endpoints were the need of noninvasive mechanical ventilation (NIMV) and invasive mechanical ventilation (IMV).

Continuous variables were expressed as mean (standard deviation) or median (interquartile ranges), according to data distribution, and were compared using the Student’s *t*-test or the Mann–Whitney U test, when appropriate; categorical variables were expressed as counts and percentages and compared using the Chi-square or Fisher’s exact tests, as appropriate. Associations between candidate variables and endpoints were assessed using both univariate and multi-variate Cox regression analyses, and hazard ratios were calculated. We have evaluated survivors compared with non-survivors and patients who needed ventilation (both invasive and non-invasive) compared with patients without ventilation.

The discriminatory power of the analyzed variables for predicting mortality was tested by means of a receiver operating characteristic (ROC) curve analysis with the area under the ROC curve (AUC) determination.

For the regression analysis, variables were dichotomized according to cut-off values derived during the data analysis for this study, using the Youden index arising from the ROC curve analysis.

For each biomarker, sensitivity, specificity, negative and positive predictive values (NPV, PPV), negative and positive likelihood ratio (LR−, LR+) and odds ratio with CI 95% were also reported for mortality, IMV and NIMV.

Kaplan–Meier curves were created to estimate the overall survival and compared using the log-rank test.

All analyses were performed with SPSS software. Tests were considered statistically significant if they yielded two-tailed *p*-values <0.05. For the multivariate analysis, we used variables resulting as statistically significant in the univariate analysis.

## 3. Results

The demographic and clinical characteristics of the study population are summarized in Table 1. The patient population had a mean age of 63 ± 14.7 years. Hypertension (40.8%), cardiovascular diseases (17.1%) and diabetes (13.1%) represented the most frequent comorbidities (Table 1)

Among the comorbidities reported, obesity did not show significant differences between survivors and non-survivors, whereas malignancy showed a statistical level close to the significance. All the other comorbidities showed a significant difference between the two groups considered. Evaluating the secondary outcomes, cardiovascular disease and malignancy did not show significant differences between IMV and no-IMV, whereas all the other comorbidities showed a significant difference. For the last group of patients, only hypertension and renal disease showed significant differences between NIMV and no-NIMV patients.

All the biomarkers evaluated just after triage showed increased values in non-survivors as compared to survivors as well as in IMV and NIMV compared to no-IMV and no-NIMV, reaching always a statistically significant level (Table 2)

Table 3 shows the results of the univariate Cox regression analysis performed to investigate the possible predictive role of clinical and demographic characteristics in patients with suspected COVID-19 infections. In addition, obesity does not seem to predict 45 days mortality, whereas malignancy showed a statistical level close to the significance in patients evaluated at the triage in the Emergency Department. All the other clinical features have shown significant odds ratio value to predict mortality in this group of patients.

Concerning the possible role in predicting need of IMV within 28 days in these patients, all the clinical features reached statistical significance except for cardiovascular disease and malignancy, whereas only hypertension and renal diseases showed a significant odds ratio value for NIMV within 28 days. All the biomarkers analyzed showed a significant role in predicting mortality, need of IMV and NIMV in patients admitted at the ED with COVID-19 infection as analyzed by the univariate Cox regression analysis (Table 3).

A multivariate analysis performed pooling together both clinical and laboratory variables, also considering the whole observation period (mortality at 45 days, IMV and NIMV at 28 days), showed that all biomarkers, except PCT, might be considered a valuable predictive tool in the mortality risk stratification at admission to the Emergency Department (Table 4).

In fact, patients with MR-proADM level higher than the cut-off value of 1105 nmol/L show an increase of mortality of almost three times (OR 2.97, IC 1.7–5.28); also, CRP levels were independently associated with a higher risk of in-hospital mortality in COVID-19 patients (OR 2.85, IC 1.73–4.69). Similarly, at ED admission, an increase of MR-proADM level was independently associated with an almost three times (OR 2.83, IC 1.49–5.36) higher risk of IMV need as well as for LDH, which showed a smaller but still significant risk (OR 2.18, IC 1.33–3.57). Only CRP showed a significant predictive value (OR 2, IC 1–3.7) for the need of NIMV.

Looking at ROC curve analysis, the prognostic ability of MR-proADM assessed at ED admissions has been shown by the good discrimination performance both for in-hospital mortality (AUC 0.85) and for prediction of IMV (AUC 0.81); it seems to be less effective, as predictive factor for NIMV prediction (AUC 0.71), with the optimal cut-off of 1105 as obtained with the Youden index. The risk stratification role of MR-proADM seems to be more powerful as compared to the other biomarkers as demonstrated by ROC curves and AUC, which resulted as significantly greater for the primary endpoint, i.e., in-hospital mortality, except for CRP (Table 5 and Figure 1).

In particular, MR-proADM showed the better PPV (65%) and especially, NPV (87%) in predicting mortality, as well as for IMV and NIMV regarding NPV (both 88%) as compared to the other biomarkers, with only CRP showing similar values (Table 5).

The good discrimination performance of MR-proADM for the primary and secondary endpoints is also shown by the survival curves (Figure 2). In fact, a higher survival rate and a reduced mechanical ventilation risk were evident for patients with values less than 1.105 nmol/L for mortality and IMV and 0.785 for NIMV at admission to the ED.

Similar results have been found for CRP (Figure 2), but with less differences when compared to MR-proADM, as shown by the narrower forks. Conversely, the performance was lower for PCT, D-dimer and LDH, as shown in Figure 3, where is evident a poor discrimination power, as shown by the narrower forks.

## 4. Discussion

The severe acute respiratory syndrome coronavirus 2 (SARS-CoV-2) infection has heavily affected the worldwide population in the last two years. Although most patients infected by SARS-CoV-2 had only a mild illness, about 5% of them suffered severe lung injury or even multiorgan dysfunction [22], requiring admission in the intensive care unit (ICU).

Consequently, Emergency Departments have seen a dramatic increase in their workload, triggering the need to optimize resources and the decision to hospitalize only seriously ill patients, to face the more adequate care level.

The utilization of biomarkers at the admission to the ED to quickly stratify risks for patients with pneumonia and other diseases has been largely reported [12,23,24]. MR-proADM has already been shown to be effective to stratify the risk in the Emergency Department for patients affected by community-acquired pneumonia (CAP) [11,25].

To our knowledge, this is the first study focused on the ability of new, as MR-proADM, and traditional biomarkers in the global risk stratification of patients with COVID-19 infections at the ED admission.

Previous studies performed with a smaller number of patients have reported that MR-pro ADM can play a role in predicting outcome in already hospitalized patients affected by COVID-19-related pneumonia [19,20].

Accordingly, we have recently shown in ICU critical patients that that MR-proADM seems to represent the most powerful biomarker for predicting death, especially when the outcome can happen earlier, within one week, thus representing a good predictor for disposition of patients from ED to ICU [26]. In addition, our recent data show that also in patients at the triage entry in the Emergency Department, MR-proADM is able to stratify the risk in terms of mortality [21].

In line with these previous studies, the median admission levels of all biomarkers checked in our studies showed significant higher values for all the endpoints considered, i.e., non-survivors vs survivors, IMV vs non-IMV and NIMV vs non-NIMV. This result suggests that these biomarkers might play a predictive role in the early risk stratification of patients with COVID-19 infections. In fact, all the biomarkers considered showed a significant predictive value for the endpoints considered when analyzed with a univariate analysis.

Considering the possible confounding effect of the demographic and clinical features of patients, a multivariate analysis was performed pooling together both the clinical characteristics and the biomarkers assessed in the study. MR-proADM showed the best predictive value for the primary endpoints and for the need of IMV, whereas it did not show significant predictive role for the need of NIMV.

In particular, it is notable that MR-proADM showed the best negative predictive value for all the endpoints considered, thus giving a relevant support to the emergency physician in the eventual decision of the patient rule-out or rule-in. Consequently, MR-proADM could determine an adequate clinical setting of patients affected by COVID-19 being able to predict also the possibility of ventilation need. This relevant information might be helpful for the emergency physician facilitating the decision-making process, thus optimizing the hospital resources. The great power of MR-proADM in the mortality risk stratification of COVID-19 patients has been further confirmed by the analysis of the ROC curves, which showed a significant greater AUC as compared to the other biomarkers analyzed.

Similarly, the survival curves showed a primary role of MR-proADM as a predictive factor in patients affected by COVID-19 in the Emergency Department. This is demonstrated by the wider fork of MR-proADM as compared to the other biomarkers, highlighting a better discrimination power for all the outcomes considered.

Our novel data are in line with previous studies in which the predictive role of MR-proADM has been evaluated in hospitalized COVID-19 patients [19,20].

Some differences among the present and previous studies need to be highlighted. First, the clinical setting. In fact, in previous studies, only critical patients admitted in the hospital wards have been enrolled. Our study, instead, has been primarily focused on patients admitted to the ED and therefore, with different degrees of impairment, ranging from asymptomatic to critical conditions. In these populations, the goal is to predict the trajectory of the illness at the onset of the symptoms and, very often, this is not easy to do.

The second important aspect of our study was the number of patients enrolled, which was considerably greater as compared to previous studies.

Similar conclusions have been reported by previous studies which have analyzed the predictive role of biomarkers in patients affected by CAP, where MR-proADM was shown to be particularly effective in stratifying the risk in patients affected by bacterial pneumonia [9,11,12]. On the other hand, PCT was particularly effective as a diagnostic tool for tailoring antibiotic therapy but with a minor impact on the predictive power [9,27,28].

Although PCR has been confirmed to be a non-specific marker of acute inflammation as it is usually influenced by many other factors [9,28,29], its predictive power, as confirmed in our study, makes it particularly useful in a context such as the Emergency Department, even for COVID-19 patients.

PCT, in line with previous studies [21,26], does not show a sufficient predictive value because, even if the ROC analysis shows a valid result, the multivariate analysis does not show a significant predictive value. This could be due to the different statistical analysis performed. In fact, in previous studies [21,26], PCT showed sufficient discriminatory power for patients who died with COVID-19 after 28 days, presumably due to complications such as bacterial superinfections. In contrast, MR-proADM was particularly effective in predicting death in patients who died rapidly, within a week, because of COVID-19.

Our novel data strengthen the role of biomarkers as a useful tool in the early risk stratification of patients presenting to the ED, especially in the age of COVID-19. We can only speculate that a diagnostic and predictive power could be enhanced by combining score and biomarkers in more complex biomarkers panels [30,31,32]. For these reasons, it is important to know how biomarkers behave in response to defined diseases such as SARS-CoV-2.

A limitation of our study is that patients were recruited in only one single hospital. Therefore, it would be desirable to extend the study to multiple centers to increase the number of enrolled patients, to confirm our results. A further limitation is represented by the retrospective design of the study.

## 5. Conclusions

This study, which to our knowledge is the first to evaluate the behavior of the MR-proADM compared to traditional biomarkers in COVID-19 patients at admission to the ED, shows that all the biomarkers utilized can help the physician in the decision-making process. Among them, MR-proADM and CRP seem to represent the most powerful biomarkers for predicting mortality and the need of ventilation in patients admitted to the Emergency Department. Therefore, the assessment of the MR-proADM level strengthens the predictive power of CRP. This is particularly useful for helping the emergency physician in the rule-in or rule-out of COVID-19 patients. Furthermore, the present study extends the previous results giving a support to the emergency physician in deciding the adequate clinical setting according to the possible need of ventilation, thus contributing to optimizing hospital resources.

However, biomarkers might oversimplify the interpretation of important variables, and consequently, they must be considered as a valid help but not as replacing clinician judgment and/or the right consideration of validated severity scores.

## Figures and Tables

**Figure 1 diagnostics-12-01971-f001:**
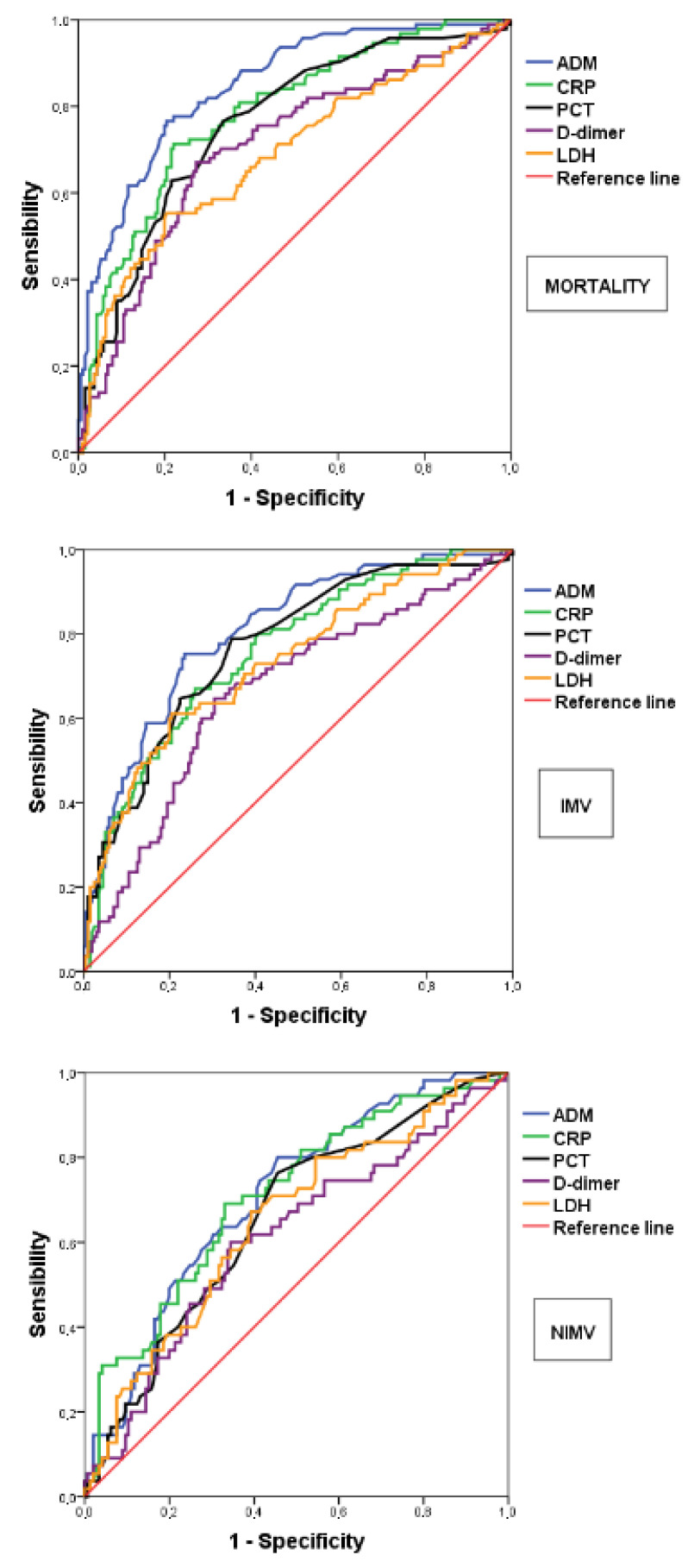
Association of candidate biomarkers with mortality and mechanical ventilation. AUROC, Area Under the Receiver Operating Characteristic curve. MR-proADM, mid-regional prodrenomedullin; CRP, C-reactive protein; PCT, procalcitonin; LDH, lactate dehydrogenase; IMV, Invasive Mechanical Ventilation; NIMV, Non-Invasive Mechanical Ventilation.

**Figure 2 diagnostics-12-01971-f002:**
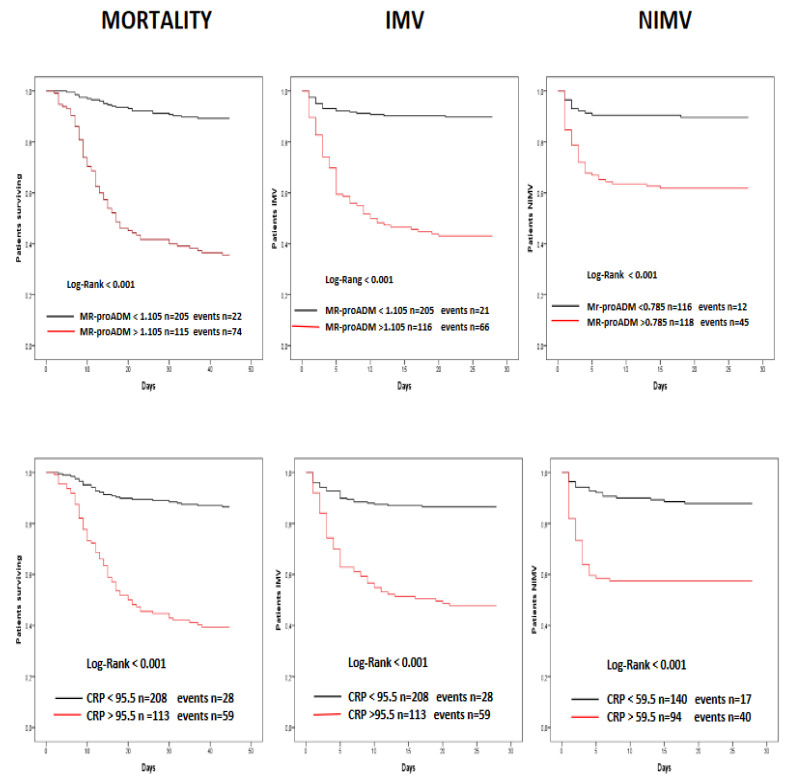
Kaplan–Meier survival curves. Stratification of patients with mid-regional proadrenomedullin (MR-proADM) levels greater or less than 1.105 nmol/L and C-reactive protein (CRP) levels greater or less than 95.5 mg/L at admission in the Emergency Department. IMV, Invasive Mechanical Ventilation; NIMV, Non-Invasive Mechanical Ventilation.

**Figure 3 diagnostics-12-01971-f003:**
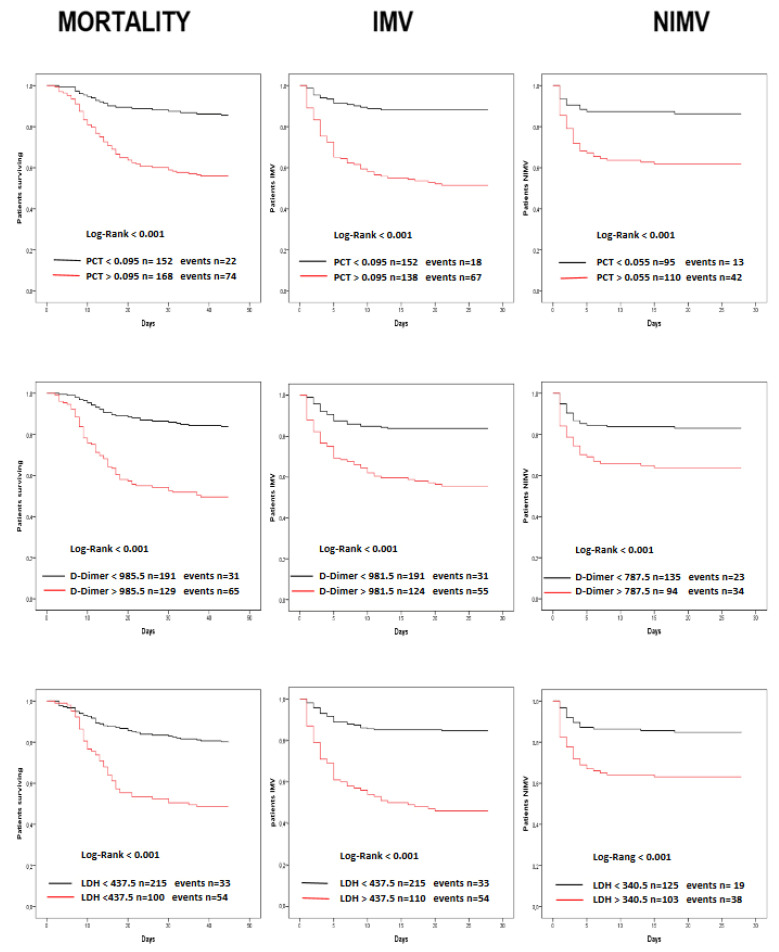
Kaplan–Meier survival curves. Risk stratification of patients with procalcitonin (PCT) levels greater or less than 0.095 ng/mL, D-dimer levels greater or less than 985.5 ng/mL and lactate dehydrogenase (LDH) greater or less than 439.5 U/L at admission in the Emergency Department. IMV, invasive mechanical ventilation; NIMV, non-invasive mechanical ventilation.

**Table 1 diagnostics-12-01971-t001:** Demographic and clinical parameters.

	Overall	Survivors	Non-Surviv	*p* Value	No-IMV	IMV	*p* Value	No-NIMV	NIMV	*p* Value
	N 321	N 224	N 97		N 234	N 87		N 177	N 57	
**Age**										
Years, mean (SD)	63.3 (14.7)	59.6 (14.6)	71.9 (11.2)	<0.001	61.4 (15.8)	68.6 (9.7)	<0.001	59.6 (16.2)	67 (12.9)	0.002
**Sex**										
Male, N (%)	215 (67.0)	145 (64.7)	70 (72.2)	0.193	146 (62.4)	69 (79.3)	0.004	107 (60.4)	39 (68.4)	0.280
Female, N (%)	106 (33.0)	79 (35.3)	27 (27.8)		88 (37.6)	18 (20.7)		70 (39.6)	18 (31.6)	
**Comorbidities**										
Hypertension, N (%)	131 (40.8)	70 (31.3)	61 (62.9)	<0.001	81 (34.6)	50 (57.5)	<0.001	51 (28.8)	30 (52.6)	0.001
Diabetes, N (%)	42 (13.1)	19 (8.5)	23 (23.7)	<0.001	21 (9.0)	21 (24.1)	<0.001	13 (7.3)	8 (14.0)	0.124
Respiratory disease, N (%)	28 (8.7)	14 (6.3)	14 (14.4)	0.017	16 (6.8)	12 (13.8)	0.050	13 (7.3)	3 (5.3)	0.588
Malignancy, N (%)	19 (5.9)	10 (4.5)	9 (9.3)	0.093	11 (4.7)	8 (9.2)	0.129	7 (4.0)	4 (7.0)	0.342
Cardiovasc. disease, N (%)	55 (17.1)	27 (12.1)	28 (28.9)	<0.001	37 (15.8)	18 (20.7)	0.303	26 (14.7)	11 (19.3)	0.407
Renal disease, N (%)	51 (15.9)	13 (5.8)	38 (39.2)	<0.001	17 (7.3)	34 (39.1)	<0.001	8 (4.5)	9 (15.8)	0.004
Obesity, N (%)	15 (4.7)	8 (3.6)	7 (7.2)	0.155	7 (3.0)	8 (9.2)	0.019	7 (4.0)	0 (0)	0.127

Values expressed in percentages (%) indicate the proportion of patients within each group for each variable. Data are presented as mean (standard deviation, SD) where specified. The chi-square (χ2) test was used to determine significance between the groups for categorical variables, Student’s t test for the variable of age. IMV, Invasive Mechanical Ventilation; NIMV, Non Invasive Mechanical Ventilation.

**Table 2 diagnostics-12-01971-t002:** Biomarkers values at triage admission.

	Overall	Survivors	Non Survivors	*p* Value	NO IMV	IMV	*p* Value	NO NIMV	NIMV	*p* Value
	N 321	N 224	N 97		N 234	N 87		N 177	N 57	
MR-proADM nmol/L										
Median	0.90	0.75	1.46	<0.001	0.79	1.42	<0.001	0.72	0.99	0.001
(Q1–Q3)	(0.63–0.33)	(0.57–1.0)	(1.14–2.37)		(0.58–1.05)	(1.11–2.14)		(0.55–0.95)	(0.80–1.30)	
CRP mg/L										
Median	61	45.90	134	<0.001	47.5	134	<0.001	35	90	<0.001
(Q1–Q3)	(24–125)	(14–86)	(72–207)		(12.0–93.0)	(68–211)		(10–75)	(48–151)	
PCT ng/mL										
Median	0.08	0.06	0.18	<0.001	0.06	0.19	<0.001	0.05	0.09	0.001
(Q1–Q3)	(0.04–0.20) N 290	(0.03–0.13) N 196	(0.10–0.40) N 94		(0.03–0.13) N 205	(0.10–0.60) N 85		(0.03–0.10) N 150	(0.06–0.20) N 55	
D-dimer ng/mL										
Median	753	647	1295	<0.001	669	1212	<0.001	603	829	0.009
(Q1–Q3)	(446–1437) N 315	(411–1063) N 219	(700–2365) N 96		(417–1148) N 229	(658–2102) N 86		(408–999) N 172	(508–1666) N 57	
LDH UI/L										
Median	349	323	456	<0.001	323	494	<0.001	303	395	<0.001
(Q1–Q3)	(268–487) N 315	(249–432) N 218	(323–597) N 97		(244–427) N 228	(343–616) N 87		(233–413) N 171	(295–500) N 57	

Data are presented as median [first quartile (Q1)-third quartile (Q3)]. The Mann Whitney U test was used to determine significance among biomarker concentrations. MR-pro ADM, mid-regional proadrenomedullin; CRP, C-reactive protein; PCT, procalcitonin; LDH, lactate dehydrogenase. IMV, Invasive Mechanical Ventilation; NIMV, Non Invasive Mechanical Ventilation.

**Table 3 diagnostics-12-01971-t003:** Univariate Cox Regression Analysis for biomarkers and clinical characteristics for the primary (survivors) and the secondary (IMV, NIMV) outcomes. Univariate Cox Regression Analysis for the prediction of 45-day mortality and 28-day IMV/NIMV.

	Cut-off	Overall (N)	Non Surv (N)	*p* Value	HR (95% CI)	Overall (N)	IMV (N)	*p* Value	HR (95% CI)	Overall (N)	NIMV (N)	*p* Value	HR (95% CI)
Age		320	96	<0.001	1.06 (1.04–1.07)	321	87	<0.001	1.03 (1.01–1.04)	234	57	0.003	1.03 (1.01–1.04)
Gender		320	96	0.182	1.35 (0.87–2.11)	321	87	0.007	2.03 (1.21–3.41)	234	57	0.301	1.34 (0.77–2.35)
Hypertension		320	96	<0.001	2.87 (1.90–4.34)	321	87	<0.001	2.16 (1.41–3.31)	231	57	0.002	2.28 (1.35–3.80)
Diabetes		320	96	<0.001	2.43 (1.51–3.92)	321	87	<0.001	2.41 (1.48–3.95)	234	57	0.137	1.76 (0.84–3.73)
Respiratory disease		320	96	0.017	2.00 (1.13–3.52)	321	87	0.038	1.91 (1.04–3.51)	234	57	0.669	0.77 (0.24–2.49)
Malignancy		320	96	0.057	1.95 (0.98–3.87)	321	87	0.115	1.80 (0.87–3.72)	234	57	0.298	1.72 (0.62–4.74)
Cardiovasc. disease		320	96	<0.001	2.47 (1.60–3.84)	321	87	0.307	1.31 (0.78–2.20)	234	57	0.385	1.34 (0.70–2.59)
Renal disease		320	96	<0.001	5.44 (3.59–8.25)	321	87	<0.001	4.85 (3.14–7.50)	234	57	0.003	2.92 (1.43–5.97)
Obesity		320	96	0.200	1.65 (0.77–3.57)	321	87	0.007	2.74 (1.32–5.67)	234	57	0.353	0.05 (0.0–29.60)
MR-proADM (nmol/L)	1.105	320	96	<0.001	9.10 (5.64–14.70)	321	87	<0.001	7.22 (4.41–11.83)	234	57	<0.001	4.20 (2.20–8.00)
CRP (mg/L)	95.5	320	96	<0.001	6.28 (4.03–9.78)	321	87	<0.001	4.79 (3.05–7.52)	234	57	<0.001	4.20 (2.40–7.50)
PCT (ng/mL)	0.095	289	93	0.001	4.62 (2.86–7.45)	290	85	<0.001	5.07 (3.01–8.54)	205	55	<0.001	3.10 (1.70–5.80)
D-dimer (ng/mL)	985	314	95	<0.001	4.18 (2.72–6.43)	315	86	<0.001	3.22 (2.08–5.01)	229	57	0.002	2.30 (1.40–4.00)
LDH (UI/L)	439.5	314	96	<0.001	3.52 (2.35–5.27)	315	87	<0.001	4.47 (2.90–6.91)	228	57	<0.001	2.80 (1.60–4.80)

HR, hazard ratio; CI, confidence interval. MR-proADM, mid-regional proadrenomedullin; CRP, C-reactive protein; PCT, procalcitonin; LDH, lactate dehydrogenase. Biomarkers cut-off values derived from ROC (Receiver Operating Characteristic) curves using the Youden index. IMV: Invasive Mechanical Ventilation; NIVM: Non Invasive Mechanical Ventilation. Cut-off changes in IMV: D-dimer 981.5 ng/mL, LDH 437.5 U/L; cut-off changes in NIMV: MR-proADM 0.785 nmol/L, CRP 59.5 mg/mL, LDH 340.5 UI/L.

**Table 4 diagnostics-12-01971-t004:** Multivariate Cox Regression Analysis pooling together biomarkers and clinical characteristics for the primary (survivors) and the secondary (IMV, NIMV) outcomes. Multivariate Cox Regression Analysis for the prediction of 45-day mortality and 28-day IMV/NIMV.

	Overall (N)	Non Surviv (N)	*p* Value	HR (95% CI)	Overall (N)	IMV (N)	*p* Value	HR (95% CI)	Overall (N)	NIMV (N)	*p* Value	HR (95% CI)
Age	284	93	0.083	1.02 (0.99–1.04)	285	85	0.386	0.99 (0.97–1.01)	200	55	0.952	0.99 (0.97–1.02)
Gender					285	85	0.095	1.63 (0.92–2.89)				
Hypertension	284	93	0.970	1.01 (0.63–1.61)	285	85	0.930	1.02 (0.64–1.64)	200	55	0.450	1.30 (0.70–2.30)
Diabetes	284	93	0.880	1.04 (0.61–1.80)	285	85	0.292	1.34 (0.78–2.31)				
Respiratory disease	284	93	0.047	1.86 (1.01–3.41)	285	85	0.248	1.49 (0.76–2.94)				
Malignancy	284	93	0.038	2.28 (1.05–4.95)								
Cardiovascular disease	284	93	0.042	1.78 (1.02–3.10)								
Renal disease	284	93	0.039	1.64 (1.02–2.62)	285	85	0.019	1.82 (1.10–3.00)	200	55	0.745	1.10 (0.50–2.40)
Obesity					285	85	0.259	1.62 (0.70–3.75)				
MR-pro ADM (nmol/L)	284	93	<0.001	2.99 (1.70–5.28)	285	85	0.001	2.83 (1.49–5.36)	200	55	0.071	2.00 (0.90–4.30)
CRP (mg/L)	284	93	<0.001	2.85 (1.73–4.69)	285	85	0.106	1.54 (0.91–2.60)	200	55	0.036	2.00 (1.0–3.70)
PCT (ng/mL)	284	93	0.602	1.17 (0.65–2.10)	285	85	0.288	1.41 (0.75–2.65)	200	55	0.075	1.80 (0.90–3.60)
D-dimer (ng/mL)	284	93	0.024	1.80 (1.08–2.99)	285	85	0.085	1.56 (0.94–2.59)	200	55	0.169	1.50 (0.80–2.80)
LDH (UI/L)	284	93	0.047	1.70 (1.01–2.84)	285	85	0.002	2.18 (1.33–3.57)	200	55	0.078	1.70 (0.90–3.10)

Age, Hpertension, Diabetes, Respiratory disease, Malignancy, Cardiovascular disease and Renal disease were used as adjusting variables within the Multivariate Cox Regression Analysis for the prediction of 45-day mortality. Age, Gender, Hpertension, Diabetes, Respiratory disease and Renal disease were used as adjusting variables within the Multivariate Cox Regression Analysis for the prediction of 28-day IMV. Age, Hypertension and Renal disease were used as adjusting variables within the Multivariate Cox Regression Analysis for the prediction of 28-day NIMV. MR-proADM, mid-regional proadrenomedullin; CRP, C-reactive protein; PCT, procalcitonin; LDH, lactate dehydrogenase.

**Table 5 diagnostics-12-01971-t005:** Prognostic accuracy of biomarkers for different outcomes.

	Outcomes	AUC (95% CI)	Cut-off	*p* Value	Sensitivity (95% CI)	Specificity (95% CI)	PPV (95% CI)	NPV (95% CI)	LR+ (95% CI)	LR- (95% CI)	OR (95% CI)
	Mortality	**0.848** (0.80–0.90)	1.105		0.77 (0.67–0.85)	0.80 (0.73–0.85)	0.65 (0.58–0.71)	0.87 (0.83–0.91)	3.75 (2.80–5.10)	0.29 (0.20–0.40)	12.76 (7.05–23.08)
MR-proADM nmol/L	IMV	**0.807** (0.75–0.86)	1.105		0.75 (0.65–0.84)	0.77 (0.70–0.82)	0.58 (0.51–0.64)	0.88 (0.83–0.91)	3.20 (2.43–4.23)	0.32 (0.22–0.47)	9.92 (5.50–17.92)
	NIMV	**0.707** (0.63–0.78)	0.785		0.80 0.67–0.90)	0.55 (0.46–0.63)	0.40 (0.35–0.45)	0.88 (0.81–0.93)	1.76 (1.41–2.19)	0.37 (0.21–0.64)	4.79 (2.29–10.0)
	Mortality	**0.785** (0.73–0.84)	95.5	0.090	0.71 (0.61–0.80)	0.78 (0.72–0.84)	0.62 (0.55–0.68)	0.85 (0.80–0.88)	3.24 (2.40–4.40)	0.37 (0.30–0.50)	8.80 (5.01–15.50)
CRP mg/L	IMV	**0.759** (0.70–0.82)	95.5	0.242	0.67 (0.56–0.77)	0.74 (0.67–0.80)	0.52 (0.45–0.59)	0.84 (0.79–0.88)	2.58 (1.95–3.40)	0.45 (0.33–0.61)	5.79 (3.34–10.06)
	NIMV	**0.709** (0.63–0.79)	59.5	0.970	0.69 (0.55–0.81)	0.67 (0.59–0.75)	0.44 (0.37–0.51)	0.85 (0.79–0.90)	2.09 (1.56–2.79)	0.46 (0.31–0.70)	4.52 (2.32–8.81)
	Mortality	**0.759** (0.70–0.82)	0.095	0.021	0.77 0.67–0.85)	0.67 (0.60–0.73)	0.53 (0.47–0.59)	0.85 (0.80–0.89)	2.29 (1.80–2.90)	0.35 (0.20–0.50)	6.50 (3.70–11.42)
PCT ng/mL	IMV	**0.769** (0.71–0.83)	0.095	0.354	0.79 (0.69–0.87)	0.66 (0.59–0.72)	0.49 (0.44–0.55)	0.88 (0.83–0.92)	2.28 (1.83–2.85)	0.32 (0.21–0.49)	7.07 (3.89–12.83)
	NIMV	**0.657** (0.57–0.74)	0.055	0.380	0.76 (0.63–0.87)	0.55 (0.46–0.63)	0.39 (0.34–0.45)	0.86 (0.79–0.91)	1.68 (1.33–2.11)	0.43 (0.26–0.71)	3.87 (1.92–7.81)
	Mortality	**0.705** (0.64–0.77)	985.5	0.001	0.67 0.57–0.76)	0.73 (0.66–0.79)	0.55 (0.48–0.61)	0.82 (0.77–0.86)	2.46 (1.90–3.20)	0.45 (0.30–0.60)	5.43 (3.18–9.28)
D-dimer ng/mL	IMV	**0.666** (0.60–0.74)	981.5	0.002	0.65 (0.54–0.75)	0.70 (0.63–0.76)	0.47 (0.41–0.54)	0.82 (0.77–0.86)	2.12 (1.63–2.76)	0.51 (0.38–0.69)	4.18 (2.44–7.15)
	NIMV	**0.610** (0.53–0.70)	787.5	0.110	0.60 (0.46–0.73)	0.66 (0.57–0.73)	0.40 (0.33–0.47)	0.81 (0.75–0.86)	1.74 (1.27–2.38)	0.61 (0.43–0.86)	2.85 (1.50–5.40)
	Mortality	**0.687** (0.62–0.76)	439.5	0.0001	0.55 (0.45–0.66)	0.80 (0.73–0.85)	0.57 (0.49–0.65)	0.78 (0.74–0.82)	2.71 (1.90–3.80)	0.56 (0.40–0.70)	4.83 (2.82–8.26)
LDH UI/L	IMV	**0.736** (0.67–0.80)	437.5	0.101	0.61 (0.50–0.72)	0.80 (0.73–0.85)	0.56 (0.48–0.64)	0.83 (0.79–0.87)	2.98 (2.16–4.11)	0.49 (0.37–0.64)	6.30 (3.61–11.00)
	NIMV	**0.649** (0.56–0.73)	340.5	0.320	0.67 (0.53–0.79)	0.61 (0.52–0.69)	0.39 (0.33–0.46)	0.83 (0.77–0.88)	1.71 (1.30–2.25)	0.54 (0.36–0.81)	3.17 (1.65–6.11)

Area Under Curve (AUC) analysis for 45-day mortality prediction and for 28-day IMV or NIMV prediction of study population. P value: differences between area of each biomarker vs MR-pro-ADM. Cut-off derived from ROC (Receiver Operating Characteristic) curves using the Youden index. CI, Confidence Interval; PPV, Positive Predictive Value; NPV, Negative Predictive Value; LR+, Positive Likelihood Ratio; LR-, Negative Likelihood Ratio; OR, Odds Ratio; IMV, Invasive Mechanical Ventilation; NIMV, Non Invasive Mechanical Ventilation. MR-proADM, mid-regional proadrenomedullin; CRP, C-reactive protein; PCT, procalcitonin; LDH, lactate dehydrogenase.

## Data Availability

Data supporting reported results are stored in databases created from LIS of the Tor Vergata University Hospital.

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
