# Peer review of "Predictive Value of MR-proADM in the Risk Stratification and in the Adequate Care Setting of COVID-19 Patients Assessed at the Triage of the Emergency Department"

_diagnostics, 2022, doi:10.3390/diagnostics12081971_

Round 1
Reviewer 1 Report
MR-proADM is a useful biomarker that remains stable in the blood and can be used in routine laboratory tests. Several papers have been reported on the relationship between SARS-CoV-2 infected patients and MR-proADM. This study was very interesting because it targeted stratifying the in-hospital mortality risk of COVID-19 patients at the triage using MR-proADM. However, no results were found for MR-proADM to exceed CRP, indicating that there is no point in measuring MR-proADM. Please add the significance of measuring ADM, not CRP.
Comments
1. How did you define the normal value for MR-proADM (<0.55 nmol/L, Line 170).
2. In Table 4, why are some items blank such as gender? Is the result reproducible even if according to sex.
3. Line 262, NIMV cut-off is 0.785 in Table 5.
4. In Table 5, why is the p-value of MR-proADM blank?
5. In general, MR-proADM levels correlate with age and is a gender difference. How about creating a multivariate ROC curve with sex age added in Figure 1?
6. Figure 2 and 3, the log-rank test showed significant differences in all items, and I could not understand what these figures intended.
7. Please enter the name of the organization that approved the ethical review board.
8. Please align the number of significant digits in Tables.
9. The resolution of Tables and Figures is low.
Author Response
MR-proADM is a useful biomarker that remains stable in the blood and can be used in routine laboratory tests. Several papers have been reported on the relationship between SARS-CoV-2 infected patients and MR-proADM. This study was very interesting because it targeted stratifying the in-hospital mortality risk of COVID-19 patients at the triage using MR-proADM. However, no results were found for MR-proADM to exceed CRP, indicating that there is no point in measuring MR-proADM. Please add the significance of measuring ADM, not CRP.
We think that every biomarker, which could be helpful to physicians for diagnosis, should be evaluated. In fact, in the Discussion (Page 14, paragraph 7 highlighted in yellow) we state “We can only speculate that a diagnostic and predictive power could be enhanced by combining score and biomarkers in more complex biomarkers panels”.
Furthermore, as we detailed in the Discussion section (Page 13, last paragraph highlighted in yellow):“…..it is notable that MR-proADM showed the best negative predictive value for all the endpoints considered thus giving a relevant support to the emergency physician in the eventual decision of the patient rule-out or rule-in. Consequently, MR-proADM could determine an adequate clinical setting of patients affected by COVID-19 being able to predict also the possibility of ventilation need.”
Moreover, our study aimed at evaluating the behavior of a new biomarker in the risk stratification at the Emergency Department. The role of CRP is widely acknowledged, even though we also know that CRP is a non-specific marker of acute inflammation. As detailed in the Introduction, we thought of emphasizing the role of MR-proADM in our study even though we are confident that also CRP can gather valuable information for the emergency physician. Consequently, according to the reviewer’s suggestion, we have modified the Discussion (page 14, paragraph 5 highlighted in yellow) as follows: “ Although CRP has been confirmed to be a non-specific marker of acute inflammation as it is usually influenced by many other factors [9,28,29], its predictive power, as confirmed in our study, makes it particularly useful in a context such as the Emergency Department, even for COVID-19 patients.”
1. How did you define the normal value for MR-proADM (<0.55 nmol/L, Line 170)?
This is the cutoff suggested by the manufacturer's procedure. Levels of MR-proADM above 0.55 nmol/L are suggestive of infection. Anyway, we changed in Methods the normality range between 0.05 and 0.55 nmol/L. Changes are highlighted in yellow. (Page 4 – 2.2 Blood sample collection, paragraph 3)
2. In Table 4, why are some items blank such as gender? Is the result reproducible even if according to sex?
In the multivariate analysis of Table 4, the blank items are not significant variables for that statistical analysis. Moreover, as shown in Table 4, gender does not seem to affect significantly our results when all the variables are analyzed together in the multivariate analysis.
3. Line 262, NIMV cut-off is 0.785 in Table 5.
The cut-off 1,105 has been obtained by the ROC curves for died and IMV patients, instead the cut-off 0,785 has been obtained for NIMV patients. We added this information at page 11, paragraph 2.
4. In Table 5, why is the p-value of MR-proADM blank?
As indicated in the legend of Table 5, the p value is related to the significance of MR-proADM vs the other biomarkers, i.e. CRP, PCT, D-dimer and LDH.
5. In general, MR-proADM levels correlate with age and is a gender difference. How about creating a multivariate ROC curve with sex age added in Figure 1?
The ROC curve for sex is not possible to do since sex is a categorical and not numerical variable. If the question refers to different ROC curves adjusted for age and sex, we believe that ROC curves with univariate analysis are useless. On the other hand, we think that to evaluate the effect of a single variable on the different outcomes correctly, the more appropriate analysis is the Cox regression that takes into account the variable “time”. Further, on the basis of our knowledge, MR-proADM levels do not correlate with age as demonstrated by the fact that the cut-off is unique not taking into account age differences.
6. Figure 2 and 3, the log-rank test showed significant differences in all items, and I could not understand what these figures intended.
In a log-rank test is important also considering the distance between the two curves, also named as fork. This distance is directly related to the the discrimination power of the variable for the outcome considered. It is evident that the fork of MR-proADM is substantially wider as compared to the other biomarkers. However, the reviewer comments allow us to explain better this concept in the manuscript. Consequently, we have changed the Results section (Page 11, paragraph 3 highlighted in yellow) as follows “Similar results have been found for CRP (Figure 2) but with less differences when compared to MR-proADM as shown by the narrower fork. Conversely, the performance was lower for PCT, D-dimer and LDH, as shown in Figure 3, where it is evident a lower discrimination power, as shown by the narrower fork.”
In addition, we have also modified the Discussion (Page 14, paragraph 1 highlighted in yellow) in order to strengthen this concept as follows “Similarly, the survival curves showed a primary role of MR-proADM as predictive factor in patients affected by COVID-19 in the Emergency Department. This is demonstrated by the wider fork of MR-proADM as compared to the other biomarkers, highlighting a better discrimination power for all the outcomes considered.”
7. Please enter the name of the organization that approved the ethical review board.
Although the name of the organization was specified in the "Institutional Review Board Statement" (page 15), we added this information in the Study Design section too (page 4).
8. Please align the number of significant digits in Tables.
Done
9. The resolution of Tables and Figures is low.
We have improved the resolution of Tables and Figures.
Reviewer 2 Report
The authors have designed a retrospective observational single-center research to study the utility of five biomarkers (MR-proADM, Procalcitonin, C-reactive protein (CRP)D-dimer and LDH as markers of mortality risk and need of mechanical ventilation in COVID19 patients admitted to ED.
MAJOR COMMENTS
The paper is too much focused on one of the five biomarkers they have studied. It is true that MR-proADM is a relatively new biomarker, but the other biomarkers, especially CRP, also showed nice results predicting mortality risk and need of mechanical ventilation. I suggest discussing the role of the five biomarkers they have studied and not only the role of MR-proADM. This is especially important for clinical practice due to the fact that majority of institutions don’t have MR-proADM available in the laboratory.
2. Title: Information of the title is limited to one of the five biomarkers. I suggest including all the biomarkers. For example, “Predictive value of MR-pro-ADM, PCT, CPR, D dimer and LDH in the risk stratification of COVID-19 patients at the Emergency Department”.
3. Blood sample collection. The authors provide normal values for CRP (< 5 mg/L), PCT (< 0.5 ng/mL) and MR-proADM (< 0.55 mmol/L). References are needed. In my opinion, these cutoff values are used for the diagnosis of bacterial infection, but there are not normal values (for example PCT normal values in healthy people are < 0.1 ng/mL).
4. It would be interesting to discuss why PCT has a prognostic accuracy for mortality and IMV similar to CRP (table 5. AUC is similar) but in the multivariate analysis PCT is the only biomarker without statistical significance for prediction of mortality and IMV.
5. Limitations: A specific item should be dedicated to the limitations of the study. There are more limitations than single-center study. The main limitation is that the study is retrospective. Other limitation is that biomarkers were measured when the patient was admitted to ED, but we don’t know the hours of evolution; kinetics of biomarkers are different and it is important for example to know the time since fever started.
MINOR COMMENTS
Figure 3. Value of log-Rank of PCT for mortality is not showed.
Conclusions should be shorter. Authors provide an item of conclusions and in the last paragraph they start with “In conclusion”. It is like the conclusion of conclusions.
Not only MR-proADM is important for the conclusion. It is important to emphasize the role of CRP.
Author Response
MAJOR COMMENTS
1. The paper is too much focused on one of the five biomarkers they have studied. It is true that MR-proADM is a relatively new biomarker, but the other biomarkers, especially CRP, also showed nice results predicting mortality risk and need of mechanical ventilation. I suggest discussing the role of the five biomarkers they have studied and not only the role of MR-proADM. This is especially important for clinical practice due to the fact that majority of institutions don’t have MR-proADM available in the laboratory.
We think that every biomarker which could be helpful to physicians for diagnosis should be evaluated. In fact, in the discussion (Page 14, 5th paragraph) we state “We can only speculate that diagnostic and predictive value could be enhanced by combining score and biomarkers in more complex biomarkers panels”
Moreover, our study was aimed at evaluating the behavior of a new biomarker in the risk stratification at the Emergency Department. The role of CRP is widely acknowledged, even though we also know that CRP “has been confirmed to be a non-specific marker of acute inflammation as it is usually influenced by many other factors”(Discussion , page 14, 4th paragraph), and for this reason it could be less helpful in a setting like the Emergency Department.
Furthermore, as we detailed in the Discussion section:“…..it is notable that MR-proADM showed the best negative predictive value for all the endpoints considered thus giving a relevant support to the emergency physician in the eventual decision of the patient rule-out or rule-in. Consequently, MR-proADM could determine an adequate clinical setting of patients affected by COVID-19 being able to predict also the possibility of ventilation need.”
As detailed in the Introduction, we thought of emphasizing the role of MR-proADM in our study even though we are confident that also CRP can gather valuable information for the emergency physician. Consequently, according to the reviewer’s suggestion, we have modified the Discussion (page 14, 4th paragraph) as follows: “ Although CRP has been confirmed to be a non-specific marker of acute inflammation as it is usually influenced by many other factors [9,28,29], its predictive power, as confirmed in our study, makes it particularly useful in a context such as the Emergency Department, even for COVID-19 patients.”
2. Title: Information of the title is limited to one of the five biomarkers. I suggest including all the biomarkers. For example, “Predictive value of MR-pro-ADM, PCT, CPR, D dimer and LDH in the risk stratification of COVID-19 patients at the Emergency Department”.
According to the above considerations we decided to maintain the original title.
3. Blood sample collection. The authors provide normal values for CRP (< 5 mg/L), PCT (< 0.5 ng/mL) and MR-proADM (< 0.55 mmol/L). References are needed. In my opinion, these cutoff values are used for the diagnosis of bacterial infection, but there are not normal values (for example PCT normal values in healthy people are < 0.1 ng/mL).
We have inserted the normality range for all biomarkers analysed. (See page 4, paragraph 5).
4. It would be interesting to discuss why PCT has a prognostic accuracy for mortality and IMV similar to CRP (table 5. AUC is similar) but in the multivariate analysis PCT is the only biomarker without statistical significance for prediction of mortality and IMV.
We thank the reviewer for the comment because it can help to clarify better the results obtained. First of all, carefully looking at table 5, PCT has an AUC value substantially lower as compared to MR-proADM, for both the outcomes, and for IMV it does not reach the statistical significance. We have also to consider that while the ROC curve analyzes the sensibility and specificity of a variable in predicting the outcome, the multivariate analysis (Cox regression) explores the ability of a variable to be correlated to an outcome related to time when we have considered the single role of all involved variables evaluated in the univariate analysis. In a previous study of our group, focused on critical patients charged in ICU, it was evident how MR-proADM has the best predictive power for patients who die in the first week, for COVID-19. On the contrary, for patients who died within 28 days, with COVID-19 but for complications including bacterial superinfections, the role of MR-proADM is less relevant while the role of PCT increases as predictive value. This could be a possible explanation for the different behavior of PCT in the statistical analysis. Anyway, following the reviewer’s suggestions, we add a paragraph in the Discussion Section (Page 14, paragraph 5) as follows: "PCT, in line with previous studies [21,26], does not show a sufficient predictive value because, even if the ROC analysis shows a valid result, the multivariate analysis does not show a significant predictive value. This could be due to the different statistical analysis performed. In fact, in previous studies [21,26], PCT showed sufficient discriminatory power for patients who died with COVID-19 after 28 days, presumably due to complications such as bacterial superinfections. In contrast, MR-proADM was particularly effective in predicting death in patients who died rapidly, within a week, because of COVID-19."
5. Limitations: A specific item should be dedicated to the limitations of the study. There are more limitations than single-center study. The main limitation is that the study is retrospective. Other limitation is that biomarkers were measured when the patient was admitted to ED, but we don’t know the hours of evolution; kinetics of biomarkers are different and it is important for example to know the time since fever started.
We thank the reviewer for the valuable suggestion that might improve the impact of the manuscript. Accordingly we have add a specific item dedicated to the limitations of the study. However, we do not consider the biomarker measurement at the admission as a limit but, on the contrary, as a strength of the study because this allows a rapid information for the emergency physician thus helping his decision making, particularly considering the rule-out or the rule-in of COVID-19 patients in the Emergency Department.
MINOR COMMENTS:
Figure 3. Value of log-Rank of PCT for mortality is not showed.
Done
Conclusions should be shorter. Authors provide an item of conclusions and in the last paragraph they start with “In conclusion”. It is like the conclusion of conclusions.
Done
Round 2
Reviewer 1 Report
The author addressed all comments.